mathematical modelling/systems theory

game theory, social modelling, class separation, social stratification

**Author for correspondence:**
Claudius Gros
e-mail: gros07@itp.uni-frankfurt.de

# Self-induced class stratification in competitive societies of agents: Nash stability in the presence of envy

## Claudius Gros

Institute for Theoretical Physics, Goethe University Frankfurt, Frankfurt am Main, Germany

 CG, 0000-0002-2126-0843

Envy, the inclination to compare rewards, can be expected to unfold when inequalities in terms of pay-off differences are generated in competitive societies. It is shown that increasing levels of envy lead inevitably to a self-induced separation into a lower and an upper class. Class stratification is Nash stable and strict, with members of the same class receiving identical rewards. Upper-class agents play exclusively pure strategies, all lower-class agents the same mixed strategy. The fraction of upper-class agents decreases progressively with larger levels of envy, until a single upper-class agent is left. Numerical simulations and a complete analytic treatment of a basic reference model, the shopping trouble model, are presented. The properties of the class-stratified society are universal and only indirectly controllable through the underlying utility function, which implies that class-stratified societies are intrinsically resistant to political control. Implications for human societies are discussed. It is pointed out that the repercussions of envy are amplified when societies become increasingly competitive.

# 1. Background

Is it possible that societies separate on their own into distinct social classes when everybody is otherwise interchangeable, born equal? This is the question examined here. Being equal means in a game-theoretical setting that agents have access to the same options and pay-off functions. Starting with random initial policies, strategies evolve according to the pay-off received on the average. For our investigation, we assume that three constituent features characterize the pay-off function. Firstly, options come with a range of distinct pay-offs. Secondly, competition for resources is present, which implies that agents selecting the same option are penalized. Thirdly, players care how they are doing with respect to others. We show that an

endogenous transition to a strictly class-stratified society takes place when these three conditions are fulfilled.

It is well established that people live not in isolation, but that social context influences memory, cognition and risk taking in general [1–3], that it leads to accountability [4] and to group decision making [5]. A key aspect of social context is the quest for social status [6,7], which has been modelled using several types of status games [8,9]. Of particular relevance to our approach is the notion that the satisfaction an individual receives from having and spending money depends not only on the absolute level of consumption, but also on how this level compares with that of others [10]. This view has seen widespread support from relative income theory [11,12]. Relative gauges are considered similarly to be of relevance for the definition of poverty [13,14].

The outcome of a game may be considered fair in a social context when nobody has an incentive to trade the reward received. For the problem of allocating multiple types of goods, which may be either divisible or indivisible, like apples, banana and kiwis, the outcome is said to be free of envy when the recipients are content with their bundles [15,16]. Here, we use envy in analogy to denote the propensity to compare rewards between agents. When relative success is important it implies that the pay-off function is functionally dependent on the outcome, the average pay-off received. A feedback loop is such established. It is well known, e.g. from the theory of phase transitions in physics [17], that feedback loops can lead to collective phenomena and hence to novel states. Indeed we find that envy induces a new state, a self-induced class-stratified Nash equilibrium.

We examine here the interaction between social context and competition for scarce resources, which lies at the core of many games. A typical example is the Hawks and Doves framework, for which the reward is divided when both agents select the same behavioural option. In a society of agents, a range of options yielding distinct pay-offs will be in general available. In this setting, competition may force agents to select different strategies, for instance, to settle for the second best course of action when the option with the highest prospective reward has already been claimed by somebody else. The outcome is a multi-agent Nash state, forced cooperation, in which agents seemingly cooperate by avoiding each other, but only because it pays off and not out of sheer good will. Other forms of cooperation [18], such as reciprocal altruism [19] and indirect reciprocity [20], share this trait. A key aspect of forced cooperation is that it is unfair in terms of reward differentials, the precondition for envy to take effect.

Forced cooperation can be argued to be a generic feature of real-world societies, both when agents are differentiated or not. In ecology, for instance, non-uniform resource allocation is observed in competitive population dynamics models when resources are scarce [21]. Envy has hence the potential to induce novel societal states in which just the initial conditions, and not differences between agents *per se*, determine in which class someone ends up. In previous studies, class structures have been presumed to exist [22], or to be dependent on as-of-birth differences [8]. Clustering into distinct classes may occur similarly for networks of agents when comparison is restricted to neighbours [23].

Outcome and input, the reward received and the structure of the pay-off function, are interdependent when envy is present, a set-up that is typical for dynamical systems studied by complex systems theory [24]. Key aspects of the present investigation, including in part terminology and analysis methods, are hence based on complex systems theory. One could also ask if it would be feasible to optimize properties of the stationary state considered desirable, such as fairness, as done within mechanism design theory [25]. An example would be to set incentives for prosocial behaviour [26], with the overall aim to optimize society [27]. This is a highly relevant programme. But what if the stationary state of the society has in part universal properties that cannot be altered by changing the underlying utility function, being independent of it? We find that this is precisely what happens when envy is relevant.

Our basic model is motivated by a shopping analogy. A clique of friends gathers for an exclusive wine tasting, with everyone shopping beforehand. There are several wine outlets, each specialized in a specific quality. In the wine cellars, a wide selection of vintage years are kept in storage, but only a single bottle per year. Shopping in the same wine cellar as somebody else implies then that someone has to be content with the second-best vintage year. At the gathering, the friends enjoy the wine, becoming envious if somebody else made the better deal. Both extensive numerical simulation of the shopping trouble model and an encompassing analytic treatment of the class-stratified state are presented. An overview of the terminology used is given in §4.5.

## 2. Model

The shopping trouble model is defined directly in terms of strategies, which are given by the probabilities $p^\alpha(q_i)$ that agent $\alpha$ selects option $i$. The quality $q_i$ corresponds to the numerical value associated with

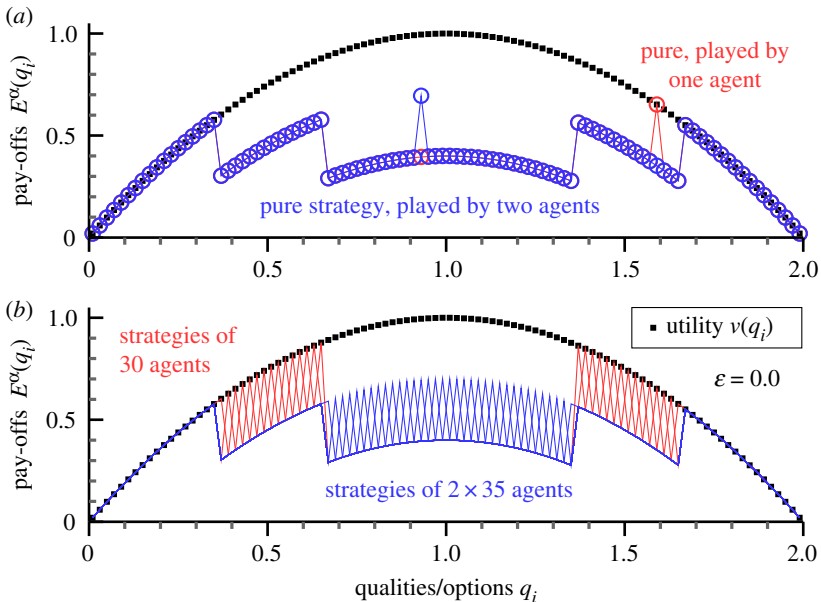

**Figure 1.** Forced cooperation. The shopping trouble model in the absence of envy, $\varepsilon = 0$. For $\kappa = 0.3$, $M = 100$ agents and $N = 100$ options, the pay-offs $E^\alpha(q_i)$ (connected by lines) obtained using evolutionary dynamics are shown. The underlying utility function, $v(q_i) = 1 - (1 - q_i)^2$, has been included as a reference (black squares). Only pure strategies are played, which implies that the rewards $R^\alpha$ correspond to the peaks of the respective pay-offs; compare (2.1). A total of 65 options are selected, 30 by a single agent (red) and 35 by two agents (blue). Competition forces the agents to accept inequalities in terms of a wide range of different rewards. (a) Two example strategies. For most qualities $q_i$, the pay-offs $E^\alpha(q_i)$ (red/blue symbols) fall on top of each other. (b) All $M$ strategies. In order to avoid overcrowding symbols are not shown.

option $i$. For convenience, we consider equidistant qualities $q_i \in [0, 2]$. The support of a strategy is given by the set of options for which $p^\alpha(q_i) > 0$. A strategy is pure when the support contains a single option, and mixed otherwise. In the model, $M$ agents have $N$ options to select from, where $N$ may be either smaller or larger than $M$.

The pay-off an agent receives when selecting $q_i$ is $E_i^\alpha$. On average, agents receive the reward $R^\alpha$,

$$R^\alpha = \sum_i p^\alpha(q_i)E_i^\alpha \quad \text{and} \quad \bar{R} = \frac{1}{M}\sum_\alpha R^\alpha, \tag{2.1}$$

where we have defined also the mean reward $\bar{R}$ of all agents. For the shopping trouble model, the pay-off $E_i^\alpha$ contains three terms

$$E_i^\alpha = v(q_i) - \kappa \sum_{\beta \neq \alpha} p^\beta(q_i) + \varepsilon\, p^\alpha(q_i) \log\left(\frac{R^\alpha}{\bar{R}}\right). \tag{2.2}$$

The first term, $v(q_i) = 1 - (1 - q_i)^2$, is the underlying utility function; compare figure 1. Its functional form, as an inverted parabola, is motivated by the shopping analogy. In this case, products having a bare utility $u(q_i)$ can be acquired at a price $q_i$ in the $i$th shop. The bare utility should be concave, in view of the law of diminishing utility [28], say $u(q_i) = a \log(q_i + 1)$. The utility entering (2.2), $v(q_i) = u(q_i) - q_i$, is in this case well approximated by an inverted parabola.

The second term in (2.2) encodes competition. A penalty $\kappa(m - 1)$ is to be paid by all $m$ agents when these $m$ agents decide on the same option. It is troublesome, in the shopping analogy, to buy something in a crowded shop. Encoding competition directly in terms of the strategy, as done in (2.2), is an adaptation of the framework used commonly for animal conflict models [29], such as the war of attrition and all pay auctions.

The third term in (2.2) encodes the desire to compare one's own success, the reward $R^\alpha$, with that which others receive. As a yardstick, the average reward $\bar{R}$ has been taken, with the envy $\varepsilon$ encoding the intensity of the comparison. The log-dependency, $\log(R^\alpha/\bar{R})$, is consistent with the Weber–Fechner Law, namely that the brain discounts sensory stimuli, numbers and time logarithmically [30–32]. Equivalent logarithmic dependencies have been found for the production of data [33], and decision-induced chart rankings [34]. For small relative deviations from the mean, when

$\delta R^\alpha = (R^\alpha - \bar{R})/\bar{R} \ll 1$, the envy term becomes linear, $\log(R^\alpha/\bar{R}) \sim \delta R^\alpha$. Envy is then directly proportional to $R^\alpha - \bar{R}$, a functionality that is equivalently at the basis of status seeking games [35]. In effect, the rationale behind the envy term is straightforward. When happy, when $\log(R^\alpha/\bar{R}) > 0$, the agent reinforces the current strategy, which is encoded by $p^\alpha(q_i)$, trying to change it instead when $\log(R^\alpha/\bar{R}) < 0$.

The utility function (2.2) of the shopping trouble model can be considered to encode status seeking, albeit indirectly. Agents try to maximize utility not only in absolute, but also in relative terms, with the envy parameter $\varepsilon$ determining the relative weight of the two contributions. Outperforming others corresponds in this interpretation to increased levels of social status. In difference to standard status seeking games [36], for which agents follow two separate objectives, utility and status maximization, the shopping trouble model contains only a single, combined utility. The issue of Pareto optimality does hence not arise.

The bare formulation of the here introduced shopping trouble game, as given by (2.2), is supplemented by the concept of migration. One postulates that agents receiving negative rewards leave the society in search of better opportunities. Better no reward at all than to engage with detrimental returns. Negative rewards appear for large $\kappa$ and elevated densities $\nu = M/N$ of agents per options, e.g. necessarily when $M = 2N$ and $\kappa > 1$. For $M < N$, there are in contrast always Nash equilibria for which all individual rewards are positive. Migration is induced additionally by the envy term, as $\log(R^\alpha/\bar{R})$ diverges negatively for $R^\alpha \to 0$. Numerically, we solved the shopping trouble model using standard replicator dynamics [37],

$$p_i^\alpha(t+1) = \frac{p_i^\alpha(t)E_i^\alpha(t)}{\sum_j p_j^\alpha(t)E_j^\alpha(t)}. \tag{2.3}$$

For a smooth convergence, one adds a constant offset $E_0$ to the pay-offs on the right-hand side. The offset helps in particular to avoid the occurrence of negative rewards, which can arise intermediately when a time evolution scheme is discrete in time, as for (2.3). Typically, we took $E_0 = 20$, iterating $5 \times 10^5$ times. A defining feature of the shopping trouble model is that all agents have functionally identical pay-offs. Only the starting strategies, which we drew from a flat distribution, differentiate between agents.

# 3. Results

In the absence of envy, when $\varepsilon = 0$, agents just need to compare the pay-off $v(q_i) - \kappa$ of options already taken by somebody else to the ones that are still available. For $\kappa = 0.3$, the outcome is presented in figure 1. Qualities with larger utilities are doubly taken, lower returning options on the other hand only by a single agent. The resulting Nash state is unique. Agents avoid each other, as far as possible, which could be interpreted as cooperation. Cooperation is, however, not voluntary, but forced by the penalty $\sim\kappa$ incurring when not cooperating. A consequence of forced cooperation is that the pay-offs received by individual agents vary considerably. This is notable, as all players start out equal, differing only with respect to their initial strategies.

The forced cooperating state is modified once $\varepsilon$ becomes finite, retaining, however, its overall character for moderate envy. Altogether two types of multi-agent Nash equilibria are observed.

— *Forced cooperation.* The distribution of rewards is continuous. Pure strategies dominate. With increasing envy, mixed strategies become more frequent. Stable for small to intermediate $\varepsilon$.
— *Class separation.* The society separates strictly into an upper and a lower class. Upper-class agents play exclusively pure strategies, all lower-class agents the identical mixed strategy. Agents belonging to the same class receive identical rewards. The number of upper-class agents decreases monotonically with increasing envy, towards one, the monarchy state. Stable for larger $\varepsilon$.

For an initial illustration, we concentrate on a small system, with $M = N = 20$, as presented in figure 2. One observes that forced cooperation dominates for $\varepsilon = 0.5$, but with some pronounced differences to the case $\varepsilon = 0$; see figure 1. The support of pure and mixed strategies, which develop for finite envy, overlap at times, which induces varied levels of competition. The supports of different mixed strategies are distinct.

## 3.1. Endogenous class stratification

A complete self-organized reorganization of the spectrum of policies is observed with raising strength of envy. The society of agents separates on its own into two distinct classes, an upper and a lower class.

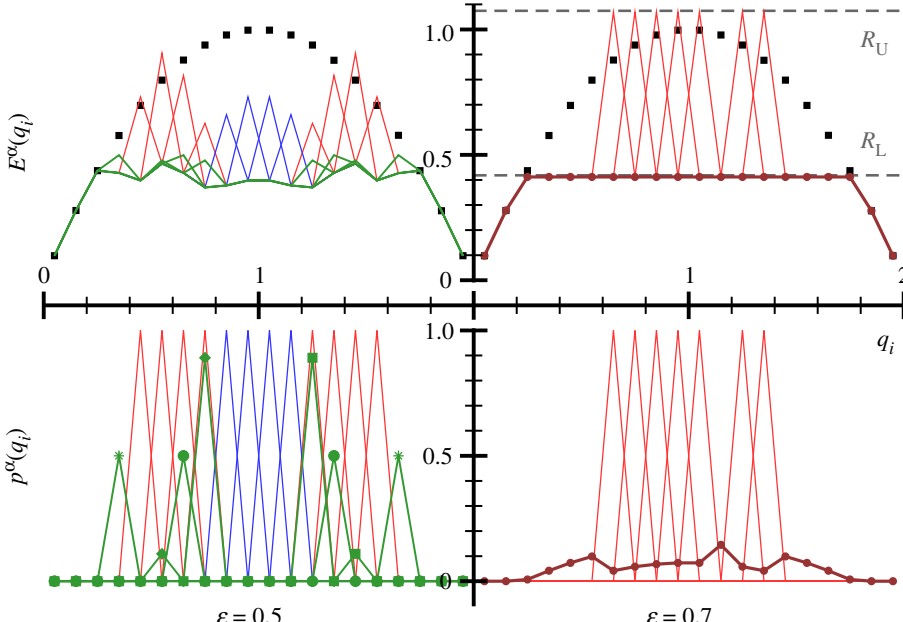

**Figure 2.** Spontaneous policy reorganization. For $\kappa = 0.3$, $M = 20$ agents and $N = 20$ options, the pay-offs $E^\alpha(q_i)$, as connected by lines (top panels), and the respective strategies $p^\alpha(q_i)$ (bottom panels). For $\varepsilon = 0.5$ (left panels) and $\varepsilon = 0.7$ (right panels). For the simulations, equation (2.3) has been used. Symbols have been added for clarity to the four individual mixed strategies in the lower-left panel. For $\varepsilon = 0.7$, seven agents play pure strategies (red), the other 13 agents the identical mixed strategy (brown). Included in the upper-right panel are the theory predictions for the lower/upper-class rewards $R_L$ and $R_U$ (dashed lines), as given by (3.1) and (3.2). Compare figure 1 for $\varepsilon = 0$.

The pay-off functions of all agents are identical, which implies that this transition, as seen in figure 2 when going from $\varepsilon = 0.5$ to $\varepsilon = 0.7$, is a collective effect. The initial state of the system determines uniquely where a given agent ends up. The class-stratified state has several conspicuous properties.

— Upper-class agents follow exclusively pure strategies, avoiding competition in most cases.
— A single mixed strategy develops, played by the entirety of lower-class agents. The support of the lower-class mixed strategy covers all upper-class pure strategies.
— Only two levels of rewards are present, one for each class.

That the pay-off function $E^\alpha(q_i)$ of the lower-class is constant on the support of the lower-class mixed strategy is a necessary condition for an evolutionary stable strategy [38]. It would be favourable to readjust the $p^\alpha(q_i)$ if this was not the case. It is also not surprising that all lower-class agents receive the same reward $R_L$, given that they play identical strategies. Highly non-trivial is, however, that the rewards of all upper-class agents coincide. This can be proven analytically, as done in the Methods section. For $R_L$, the expression

$$R_L = \varepsilon \, \frac{1 - f_L}{e^{\kappa/\varepsilon} - 1} \log\left(\frac{e^{\kappa/\varepsilon} - f_L}{1 - f_L}\right) \tag{3.1}$$

is exact when upper-class policies are unique, *viz.* if no option is taken by more than one agent, which holds for most instances. The only free parameter in (3.1) is the fraction of lower-class agents, $f_L$, which needs to be determined numerically. For the Nash state shown in figure 2, one finds $f_L = 13/20$ for $\varepsilon = 0.7$. The resulting prediction (3.1) for the reward $R_L$ of the lower class agrees remarkably well with numerics, as seen in figure 2. The analytic prediction for the reward of the upper-class, $R_U$, is

$$R_U = \varepsilon \, \frac{1 - f_L e^{-\kappa/\varepsilon}}{1 - e^{-\kappa/\varepsilon}} \log\left(\frac{e^{\kappa/\varepsilon} - f_L}{1 - f_L}\right), \tag{3.2}$$

as derived in the Methods section. Again, theory and numerical simulations are in agreement.

The underlying utility $v(q_i)$ enters the theory expressions for $R_L$ and $R_U$ only implicitly, via the fraction $f_L$ of lower-class agents, but not explicitly. The properties of class-stratified states with the same $\kappa$, $\varepsilon$ and $f_L$ are hence identical and independent of the shape of the utility function. We tested this proposition performing simulations using a triangular utility function, $v(q_i) = 1 - |1 - q_i|$, finding that (3.1) and

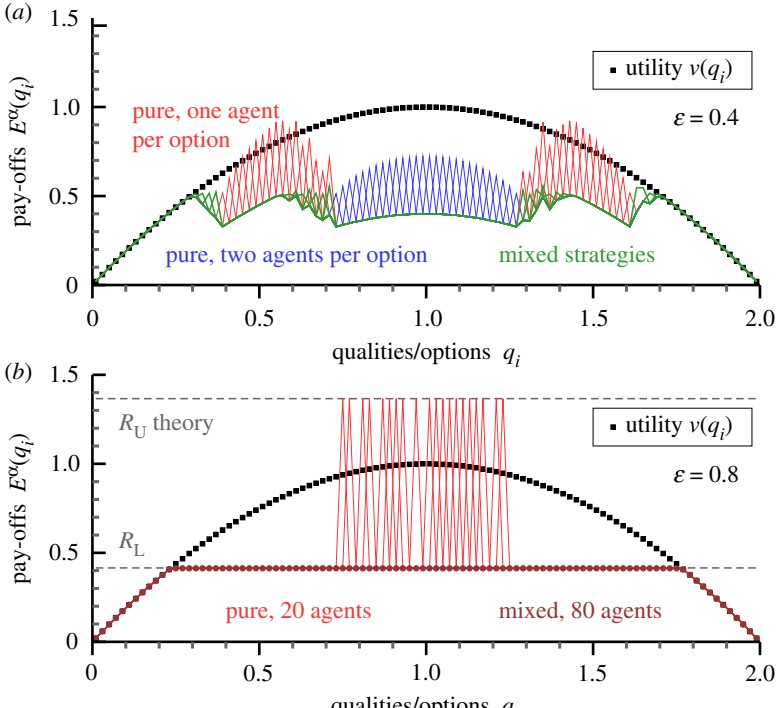

**Figure 3.** Self-induced class separation. For $M = N = 100$ and $\kappa = 0.3$, the pay-offs $E^{\alpha}(q_i)$ for the two types of Nash equilibria found, forced cooperation, and class separation. (*a*) For $\varepsilon = 0.4$. The number of agents playing mixed (green) and pure (red/blue) strategies are 10 and 90. A total of 62 pure qualities are selected, 34/28 by one/two agents. (*b*) For $\varepsilon = 0.8$. Pure strategies (red) are played by the 20 upper-class agents, with all 80 lower-class agents using the identical mixed strategy (brown). Also shown are the universal theory predictions (3.1) and (3.2) for the lower/upper-class rewards $R_L$ and $R_U$ (dashed lines). The reorganization of the pay-off spectrum is a collective effect.

(3.2) hold perfectly. Class-stratification leads as a consequence to a Nash state with universal properties, the telltale sign of a collective effect.

That upper-class agents receive identical pay-offs is an interesting aspect of universality. It is possible because the lower-class agents adapt their strategies such that the functional dependence of $v(q_i)$ on the qualities is exactly compensated by the competition term $\sim \kappa$. Evidence for this mechanism can be seen in figure 2 for $\kappa = 0.7$.

The transition from forced cooperation to a stratified society is found for all system sizes. In figure 3, we present to this end simulations for $M = N = 100$. Below the transition, here for $\kappa = 0.4$, one observes that individual mixed strategies start out at the fringes of the support regions of the pure strategies. For $\varepsilon = 0.8$, a stratified society is present, with the vast majority of agents, 80%, playing one and the same mixed strategy. These agents form the lower-class.

## 3.2. Lower-class mixed strategy

When forced cooperation is present, qualities $q_i$ with high utilities $v(q_i)$ are selected without exception by agents playing pure strategies. This is not the case for the spectrum of upper-class policies, which may have gaps in the class-stratified state, as evident both in figures 2 and 3. At first sight, this could seem a contradiction to Nash stability. Lower-class agents are, however, more likely to visit a quality $q_i$ not selected by the upper-class, as can be seen in figure 2, which leads to a competition block proportional to the competition penalty $\kappa$. It is hence not favourable for upper-class agents to switch. The occurrence of gaps implies in particular that the Nash state is not unique.

The lower-class mixed strategy has a well-defined functional form, $p_{\mathrm{mix}}(q_i) = p^{\alpha}(q_i)$, in the limit $f_U \to 0$, namely

$$
\left.
\begin{aligned}
p_{\mathrm{mix}}(q_i) &= \frac{1}{\kappa(M-1)}\left[v(q_i) - E_c\right] \\
\text{and} \quad E_c &= 1 - \left(\frac{3\kappa}{2}\right)^{2/3}\left(\frac{M-1}{N}\right)^{2/3},
\end{aligned}
\right\}
\tag{3.3}
$$

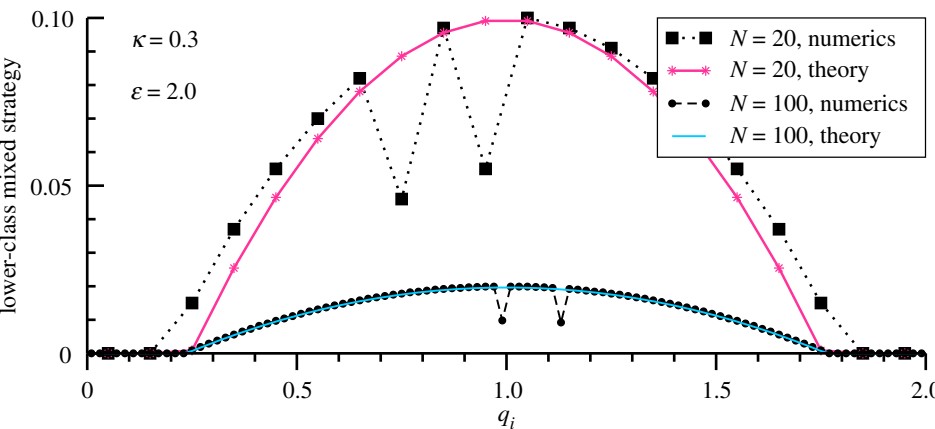

**Figure 4.** Mixed lower-class strategy. For $\kappa = 0.3$ and $\varepsilon = 2.0$, the mixed strategy $p^\alpha(q_i)$ identical to all lower-class agents. Shown are the results of numerical simulations (black) and the respective large-$N$ theory prediction (3.3). The dips in the numerical result are due to the presence of two upper-class agents (not shown), respectively, for both $M = N = 20$ and $M = N = 100$. Compare figure 2. The improved agreement between simulations and theory between $N = 20$ and $N = 100$ is consistent with the precondition of the theory, which becomes asymptotically exact in the large-$N$ limit.

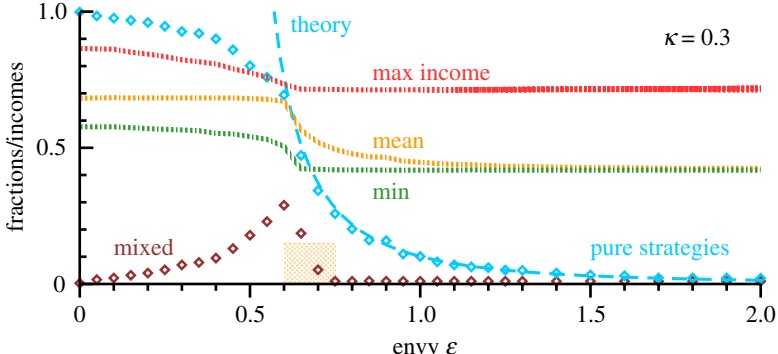

**Figure 5.** Characteristics of the Nash state as a function of envy. For $\kappa = 0.3$ and $M = N = 100$ the properties of the evolutionary stable stationary state. Numerical results are averaged over at least 10 random initial conditions. Shown is the number of distinct mixed strategies per agent (brown) and the percentage of agents playing pure strategies (blue). Denoted by 'theory' is the estimate for the fraction of upper-class agents, as determined by the self-consistency condition (3.4), which becomes exact for large $\varepsilon$. Also included are the maximal (red), the mean (yellow) and the minimal (green) monetary incomes; compare (3.5). Forced cooperation and class separation are fully present, respectively, for $\varepsilon < 0.6$ and $\varepsilon > 0.75$. In between (shaded orange), both states may be reached when starting from random initial policies.

an expression derived in the Methods section. The mixed strategy given by (3.3) is clearly non-universal, being linear in the utility $v(q_i)$, in contrast to the lower- and upper-class rewards. The expression for the reward $E_c$ is specific to the utility function $v(q_i) = 1 - (1 - q_i)^2$, and hence equally non-universal.

The simulations presented in figure 4 for $M = N = 20$ and $M = N = 100$ show that (3.3) approximates the data well when the system is large. Of interest are in particular the dips in the data for the mixed strategy, which occur for qualities $q_i$ selected by upper-class agents. These dips are essential for attaining universal $R_L$ and $R_U$, as laid out in the Methods section. Note that $p_{\text{mix}}(q_i)$ is normalized, $\sum_i p_{\text{mix}}(q_i) = 1$, when the qualities are dense.

## 3.3. Monarchy versus communism

We performed numerical simulations for a wide range of parameters, mostly for $N = 100$ options. In order to check for finite-size effects, we compared selected parameter settings with simulations for $N = 500$, finding only minor effects. In figure 5, representative data for $\kappa = 0.3$ and $M = N = 100$ are presented. Varying the filling fraction $M/N$ and/or $\kappa$ shifts the locus of the transition, leading otherwise only to quantitative changes. We define with $N_{\text{pure}}$ the number of agents playing pure strategies and with $N_{\text{mix}}$ the number of distinct mixed strategies.

For small $\varepsilon$, the fraction of mixed strategies $N_{\mathrm{mix}}/M$ raises monotonically, as shown in figure 5, attaining a maximum when the transition from forced cooperation to class stratification starts to take place, here at $\varepsilon \approx 0.6$. The width of the transition is finite, in the sense that either state may be reached when starting from random initial conditions. As a test, we ran twenty independent simulations for $\kappa = 0.3$ and $\varepsilon = 0.65$, finding that about half led to forced cooperation and half to class separation. Overall, at least 10 random initial strategies have been used for the individual data points presented in figure 5. The transition to class stratification is completed when the number of mixed strategies drops to one, which is the case for $\kappa = 0.3$ for $\varepsilon \approx 0.75$.

The fraction of agents $N_{\mathrm{pure}}/M$ playing pure strategies decreases monotonically for all $\varepsilon$, with the decrease accelerating in the transition region from forced cooperation to class separation. For large values of envy, roughly for $\varepsilon > 2.2$ a monarchy state is reached. The number of upper-class agents is now minimal, mostly just one, occasionally also two. An alternative to monarchy would be communism, namely that the entire society of agents adopts $p_{\mathrm{mix}}(q_i)$, as defined by (3.3). We find communism never to be stable, both when starting with random initial condition and when starting close to the communist state. For the latter, we performed simulations for which the initial strategies of all agents was $p_{\mathrm{mix}}(q_i)$, plus a perturbation consisting of 1% relative noise.

Included in figure 5 is an approximate analytic prediction for the fraction $f_{\mathrm{U}} = 1 - f_{\mathrm{L}}$ of upper-class agents, which is obtained from solving

$$1 - \left(\frac{3\kappa(M-1)}{2N}\right)^{2/3} = \varepsilon \, \frac{1 - f_{\mathrm{L}}}{\mathrm{e}^{\kappa/\varepsilon} - 1} \log\left(\frac{\mathrm{e}^{\kappa/\varepsilon} - f_{\mathrm{L}}}{1 - f_{\mathrm{L}}}\right), \tag{3.4}$$

self-consistently for $f_{\mathrm{L}}$. The derivation of (3.4), which is valid in the class-separated state for large $\varepsilon$, $N$ and $M$, is given in the Methods section. The agreement with the numerical results for the fraction of agents playing pure strategies, which coincides with the fraction of upper-class agents, is remarkable. Of interest is in particular the observation that the theory and numerics stay close down to the transition to forced cooperation.

## 3.4. Reward versus real-world income

The three terms entering the pay-off function (2.2) of the shopping trouble model are distinct in character. The underlying utility $v(q_i)$ and the penalty arsing from competition, $\sim\kappa$, are real-world monetary pay-off terms. Envy, the propensity to compare one's own success with that of others, could be classified in contrast as being a predominately psychological component. Taking this view, we define with

$$I^{\alpha} = R^{\alpha} - \varepsilon \log\left(\frac{R^{\alpha}}{\bar{R}}\right) \sum_i \left[p^{\alpha}(q_i)\right]^2, \tag{3.5}$$

the monetary income $I^{\alpha}$ of agent $\alpha$ as the average pay-off minus the envy term. In the forced cooperation phase, the average income $\bar{I}$ in nearly constant as a function of envy, as shown in figure 5, dropping, however, substantially once class stratification sets in. In this respect, the society is better off at low to moderated levels of envy. Class separation does not help the general public. Also included in figure 5 are the minimal and maximal incomes, $I_{\min} = \min_\alpha I^{\alpha}$ and $I_{\max} = \max_\alpha I^{\alpha}$. In the class-stratified phase, $I_{\min}$ and $I_{\max}$ correspond, respectively, to the income of the lower and of the upper-class. Both are flat, which implies that the drop of the mean monetary return $\bar{I}$ with increasing envy is due to the simultaneously occurring decrease in number of upper-class agents.

At no stage are incomes increased when envy is present in a competitive society. This result holds for the maximal, the minimal and the mean income, as evident from the data presented in figure 5. It is also conspicuous that the monetary returns of both the lower and the upper-class are essentially unaffected by the value of $\varepsilon$, once class separation sets in. This result is in agreement with the observation that the lower-class mixed strategy is well approximated by the large $\varepsilon$ limit, as shown in figure 4. Note that the reward of the upper-class diverges for $\varepsilon \to \infty$, in contrast to the monetary income. Of interest is also that $\bar{I}$ remains flat during forced cooperation, despite the fact that both $I_{\min}$ and $I_{\max}$ drop. This is due to the ongoing reorganization of the pay-off spectrum.

## 3.5. Phase diagram

In figure 6, the phase diagram as a function of $\kappa$ and $\varepsilon$, competition and envy, is presented. Systems with $N = 100$ options and $M = 50/100/150$ agents have been simulated numerically. For the onset of the transition from forced cooperation to class separation, the maximum of the number of mixed

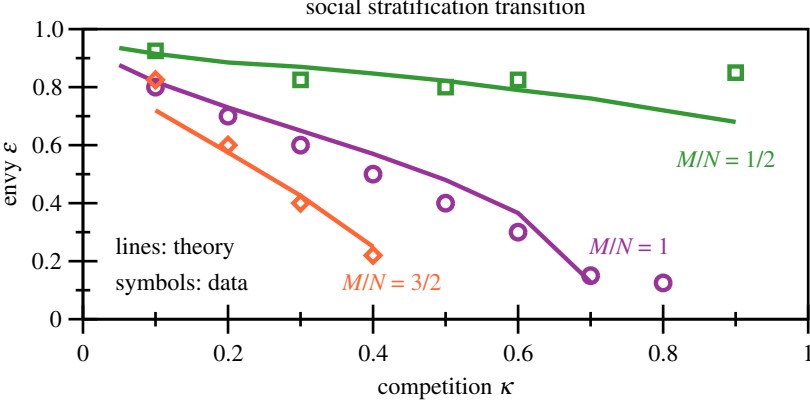

**Figure 6.** Social stratification transition. Phase diagram for the transition from forced cooperation (below the lines/symbols) to the class-separated state (above the lines/symbols). For the theory, the locus $f_U = 1/2$, as evaluated by solving (3.4) self-consistently, has been used as an indicator. These data have been obtained for $N = 100$ and, respectively, $M = 50/100/150$. In this case, the peaks in the number of mixed strategies have been taken as proxies for the transition. Compare figure 5. It is noticeable that theory and numerics track each other, in order of magnitude. The actual width of the transition has not been estimated.

strategies has been taken as an indicator. The level of envy needed for the society to phase separate decreases with increasing competition, a somewhat intuitive result. The same trend holds when increasing the density $M/N$ of agents per available options, which makes it more difficult to avoid each other.

Included in figure 6 are estimates obtained by solving the self-consistency condition (3.4) for $f_U = 1/2$. This rough estimate for the transition to class stratification tracks the numerical results surprisingly well. Deviations are seen in particular for larger $\kappa$.

# 4. Methods

The penalty term of the shopping trouble model (2.2) can be written as

$$\sum_{\beta \neq \alpha} p^\beta(q_i) = M\bar{p}(q_i) - p^\alpha(q_i) \quad \text{and} \quad \bar{p}(q_i) = \frac{1}{M}\sum_\beta p^\beta(q_i), \tag{4.1}$$

which demonstrates that agents interact via a quality-dependent mean field, the average strategy $\bar{p}(q_i)$. The number of terms is $N - 1$, which could give the impression that the shopping trouble model is not size consistent. This is, however, not the case, as both pure and mixed strategies contribute on average of the order of $1/N$ to the sum of the $\kappa$-term. For a fixed occupation density $v = M/N$, the thermodynamic limit $M, N \to \infty$ is therefore well defined. Numerically, we find that the properties of the Nash equilibria change only in minor ways when increasing $N$, but retaining $\kappa$, $\varepsilon$ and $v$.

## 4.1. Theory for the class-separated state

In the shopping trouble model, agents have functionally identical pay-off functions, which implies that *a priori* distinctions between agents are not present. Where an agent ends up, in the lower or in the upper-class, depends as a consequence solely on the respective initial conditions. For our analysis of the class-separated state, we denote with $q_U$ and $q_{\neg U}$ qualities within the support of the lower-class that are taken/not taken by upper-class agents. $M_L/M_U$ are, respectively, the number of lower/upper-class agents. We assume that upper-class agents are forced to cooperate fully, playing distinct options. Numerically, this holds in most cases.

Central to our considerations are the properties of evolutionary stable strategies [38], in particular that the pay-off is constant within the support. For the mixed strategy of the lower-class, $p_L(q_i)$, this implies that the pay-off function $E(q_i)$ is identical to the reward $R_L$,

$$R_L = v(q_{\neg U}) - \kappa[M_L - 1]p_L(q_{\neg U}) + \varepsilon p_L(q_{\neg U}) \log(R_L/\bar{R}). \tag{4.2}$$

Playing against an evolutionary stable strategy entails to receive the same constant pay-off [38]. Outside the support of their own pure strategies, upper-class agents play against the lower-class and against other

upper-class agents. For $q_i$ for which $p_L(q_i) > 0$ and for which all upper-class policies vanish, $p_U(q_i) = 0$, the consequence is that $E(q_i) = R_L$ also for upper-class agents,

$$R_L = v(q_{\neg U}) - \kappa M_L\, p_L(q_{\neg U}).$$

Numerically small deviations from can occur. The reason is that the above relation enters the evolution equation (2.3) multiplied by $p_U(q_{\neg U})$, which vanishes in the final state. The condition, that playing against an evolutionary stable strategy must yield the pay-off of the strategy in question, can be enforced consequently only while $p_U(q_{\neg U})$ is still finite. This is the case only during a transition period, while iterating towards stationarity.

Equating the two relations for $R_L$ derived so far and dividing by $p_L(q_{\neg U})$, which is positive within the support of the lower-class, yields the universal relation

$$\log\left(\frac{R_L}{\bar{R}}\right) = -\frac{\kappa}{\varepsilon} \quad \text{and} \quad R_L = \bar{R}\, e^{-\kappa/\varepsilon}, \tag{4.3}$$

which we verified numerically. Note that (4.3) is independent of the underlying utility function $v(q_i)$, of $N$ and of $M$. Denoting with $f_U = M_U/M$ and $f_L = M_L/M$ the relative fractions of upper- and lower-class agents, one has

$$\bar{R} = f_U R_U + f_L R_L \quad \text{and} \quad R_U = \frac{\bar{R}}{f_U}(1 - f_L e^{-\kappa/\varepsilon}) \tag{4.4}$$

for $R_U$, when using (4.3) for $R_L$. For (4.4) we assumed that the rewards $R^\alpha$ for upper-class agents are all identical, which we will prove shortly. Together, one finds

$$R_U - R_L = \frac{\bar{R}}{f_U}(1 - e^{-\kappa/\varepsilon}), \tag{4.5}$$

for the gap in the rewards received by the upper and the lower-class.

The two relations for $R_L$ derived so far are conditions for the $q_{\neg U}$, that is for options not taken by upper-class agents. When playing an option $q_U$ occupied by an upper-class agent, the lower-class pay-off function reads

$$\begin{aligned} R_L &= v(q_U) - \kappa\big[(M_L - 1)\,p_L(q_U) + 1\big] + \varepsilon\, p_L(q_U)\log(R_L/\bar{R}) \\ &= v(q_U) - \kappa M_L\, p_L(q_U) - \kappa, \end{aligned} \tag{4.6}$$

when using (4.3) and the precondition that there is exactly one upper-class agent with $p_U(q_U) = 1$. Note that pay-off and reward coincide for lower-class agents. We now turn to the pay-off of upper-class agents,

$$E_U^\alpha = v(q_U) - \kappa M_L\, p_L(q_U) + \varepsilon \log(R_U^\alpha/\bar{R}). \tag{4.7}$$

Here, $p_U(q_U) = 1$ has been used. With (4.6) one obtains

$$E_U^\alpha - R_L = \kappa + \varepsilon \log(R_U^\alpha/\bar{R}), \quad E_U^\alpha \to R_U, \tag{4.8}$$

which is manifestly independent of $q_U$, and hence also of the bare utility function $v(q_i)$. The independency of (4.8) with respect to the utility function implies that the pay-offs of all upper-class agents coincide, namely that $E_U^\alpha \equiv R_U$. Equating (4.5) with (4.8) yields

$$R_U - R_L = \frac{\bar{R}}{f_U}(1 - e^{-\kappa/\varepsilon}) = \kappa + \varepsilon \log\left(\frac{R_U}{\bar{R}}\right), \tag{4.9}$$

and hence

$$\begin{aligned} \frac{\bar{R}}{f_U}(1 - e^{-\kappa/\varepsilon}) &= \kappa + \varepsilon \log\left(\frac{1 - f_L e^{-\kappa/\varepsilon}}{f_U}\right) \\ &= \varepsilon \log\left(\frac{e^{\kappa/\varepsilon} - f_L}{f_U}\right), \end{aligned} \tag{4.10}$$

when using (4.4) to eliminate $R_U/\bar{R}$ on the right-hand side of (4.9). With

$$\bar{R} = \varepsilon\, \frac{1 - f_L}{1 - e^{-\kappa/\varepsilon}} \log\left(\frac{e^{\kappa/\varepsilon} - f_L}{1 - f_L}\right), \tag{4.11}$$

we obtain a universal relation for the mean reward $\bar{R}$. It follows, as the argument of the logarithm is larger than unity, that $\bar{R}$ is strictly positive. The mean reward depends only implicitly on the utility

function $v(q_i)$, through the fraction $f_L$ of lower-class agents, but not explicitly. Together with (4.8) and (4.5) the lower- and upper-class rewards $R_L$ and $R_U$ are determined as (3.1) and (3.2).

## 4.2. Identical strategies

For the case that all $M$ agents play the identical strategy $p(q_i) \equiv p^\alpha(q_i)$, the expected pay-off $E_i \equiv E_i^\alpha$ is

$$E_i = v(q_i) - \kappa(M-1)p(q_i) \rightarrow E_c. \tag{4.12}$$

With $p(q_i)$ being evolutionary stable, the pay-off $E_i$ is constant on the support, $E_i \equiv E_c$. For qualities outside the support, the pay-off $E_i$ will be lower [38]. The probability $p(q_i)$ to select an option enters $E_i$ explicitly, which implies that $p(q_i)$ is obtained by a direct inversion of (4.12). One has therefore

$$p(q_i) = \frac{1}{\kappa(M-1)}\left[v(q_i) - E_c\right]. \tag{4.13}$$

The maxima of the probability distribution $p(q_i)$ and of the utility function $v(q_i)$ coincide. The final pay-off $E_c$ is a free parameter which is determined by the normalization condition

$$1 = \sum_{i;\,p(q_i)>0} p(q_i), \quad p(q_i) \rightarrow p_i(E_c), \tag{4.14}$$

where the sum runs over the support of the policy. For finite $N$ the normalization condition (4.14) needs to be solved numerically via (2.3). Results are shown in figure 4.

## 4.3. Large numbers of options

The normalization condition (4.14) reduces to an integral for large numbers of qualities $N$. The boundary of the support is determined by

$$v(q) = E_c \quad q_\pm = 1 \pm \sqrt{1 - E_c}, \tag{4.15}$$

since $v(q) = 1 - (1-q)^2$. The normalization condition (4.14) takes then the form

$$\kappa(M-1) = \int_{-\sqrt{1-E_c}}^{\sqrt{1-E_c}} \left(1 - x^2 - E_c\right)\frac{dx}{\Delta x}, \tag{4.16}$$

when using $x = 1 - q$ and $\Delta x = 2/N$. We obtain

$$2\kappa\frac{M-1}{N} = \left(2 - \frac{2}{3}\right)(1 - E_c)^{3/2}, \tag{4.17}$$

which yields (3.3), or

$$1 - E_c = \left(\frac{3\kappa}{2}\right)^{2/3}\left(\frac{M-1}{N}\right)^{2/3}. \tag{4.18}$$

The resulting mixed strategy $p(q_i)$, as given (4.13), is in excellent agreement with simulations when only a few upper-class agents are left, as illustrated in figure 4 for $\kappa = 0.3$ and $\varepsilon = 2$.

Migration occurs when $E_c \rightarrow 0$, that is when

$$1 = \frac{3\kappa}{2}\frac{M-1}{N} \approx \frac{3\kappa\nu}{2}, \quad \nu = \frac{M}{N}, \tag{4.19}$$

where the last approximation holds for large $M$ and $N$. The carrying capacity of the society, the maximal possible density $\nu$ of agents, scales hence inversely with the strength of the competition, $\kappa$. It is independent of $\varepsilon$.

The fraction $f_U$ of upper-class agents is small when envy is large. In this limit one can approximate the lower-class reward $R_L$ with $E_c$, as determined by (4.18). This approximation, $E_c \approx R_L$, leads to (3.4), when taking also (3.1) for $R_L$ into account.

## 4.4. Scaling for two agents

The result for identical strategies, equation (4.18), has a well-defined large-$N$ limit for a constant filling fraction $\nu = M/N$. It is also of interest to consider the case of finite numbers of agents, $M$, say $M = 2$.

In the limit $N \to \infty$, the policy $p(q)$ converges in this case towards a pure strategy, *viz.* to a delta-function. The scaling for the maximum $p_{\max}$ and the width $\Delta q$ are

$$p_{\max} \sim 1 - E_{\mathrm{c}} \sim \left(\frac{1}{N}\right)^{2/3} \quad \text{and} \quad \Delta q \sim \sqrt{1 - E_{\mathrm{c}}} \sim \left(\frac{1}{N}\right)^{1/3}, \tag{4.20}$$

see (4.13) and (4.15). The width $\Delta q$ of the support shrinks only slowly when increasing the number $N$ of options, remaining substantial even for large numbers, such as $N = 10^3$. This is a quite non-trivial result, as one may have expected that the effect of competition between agents decreases faster, namely as $1/N$. Note that the scaling of the area, $p_{\max}\Delta q \sim 1/N$, is determined by the density of options, which is $N/2$.

## 4.5. Terminology

For convenience, we present here an overview of the terminology used, including for completeness selected key game-theoretical definitions. It follows that the shopping trouble model is a probabilistic competitive evolutionary game based on undifferentiated but distinguishable agents.

### 4.5.1. Options/qualities

An option is a possible course of action, like going to a shop to buy something. For a game with a large number of options, as considered here, it is convenient to associate a numerical value to an option. One may either identify the option with its numerical value, as it is usual, e.g. for the war of attrition, or distinguish them on a formal level, as done here. For an option $i$, we denote with the quality $q_i$ the associated numerical value.

### 4.5.2. Pure/mixed strategies

In simple games, like the Hawk and Dove competition, options and strategies are often not distinguished. Selecting an option, to fight or not to fight, is then identical to the strategy. On a general level, strategies define how and when a player selects one of the possible options. A strategy is pure when the agent plays the identical option at all times, and mixed otherwise.

### 4.5.3. Probabilistic game

For probabilistic games, strategies are defined in terms of probabilities. This is the case for the shopping trouble model, where $p^\alpha(q_i)$ defines the probability that agent $\alpha$ selects at any time the quality $q_i$ associated with the option $i$.

### 4.5.4. Support

A probabilistic strategy assigns a probability $p^\alpha(q_i) \geq 0$ to all possible options. One often finds that the $p^\alpha(q_j)$ are finite only for a subset of options, the support of the strategy. The size of the support is larger than one for mixed strategies, and exactly one for pure strategies.

### 4.5.5. Undifferentiated distinguishable agents

Agents are differentiated when every agent is characterized by an individual set of parameters, and undifferentiated when the same set of parameters applies to everybody. Strategies are specific to individual agents, in any case, when they are distinguishable. Indistinguishable agents share in contrast strategies.

### 4.5.6. Pay-off/reward

The pay-off function is a real-valued function of the qualities/options. The aim is to optimize the strategy such that the average pay-off is maximized. For the average pay-off the term reward is used throughout this study.

### 4.5.7. Evolutionary game

Evolutionary games are played not just once, but over and over again. After each turn, agents update their individual strategies according to the pay-offs received when selecting option $i$ with the probability $p^\alpha(q_i)$.

### 4.5.8. Competitive/cooperative game

In cooperative games, parties may coordinate their individual strategies, e.g. in order to optimize collective pay-offs. Contracts (like I select option A if you go for B) are, on the other hand, not possible for competitive games. Also possible are coalition formation or hedonic games focusing on the formation of subgroups.

### 4.5.9. Nash equilibrium

For competitive games, an equilibrium in terms of the individual strategies may be attained. In this state, the Nash equilibrium, rewards diminish when individual players attempt to change their strategies. More than one Nash state can exist for identical parameter settings. Nash stable configurations of strategies correspond to locally stable fix-points of the replicator dynamics (2.3) for evolutionary games.

### 4.5.10. Collective effect

In complex systems theory, a collective effect is present when the interaction of an extended number of constituent elements gives rise to a new type of state. An example from psychology is the emergence of mass psychology from individual behaviours. In the shopping trouble model, the transition from forced cooperation to class stratification is a collective phenomenon.

### 4.5.11. Forced cooperation

Agents may agree to select different options in cooperative games, for example in order to optimize overall welfare. Players may, on the other hand, be forced to avoid each other in competitive games, because of the penalties that would incur otherwise on individual levels. To an outside observer the resulting state has the traits of cooperation, which is in this case, however autonomously enforced.

### 4.5.12. Envy

In the context of the present study, envy is defined in terms of the pay-off function. For this, the pay-off a given agent $\alpha$ receives, when selecting a certain option $i$, depends expressively on the rewards of the other agents. Envy adds a non-monetary contribution to the reward of the player, which is positive/negative if the overall reward of the player is larger/lower than that of others.

## 5. Discussion and conclusion

The process of class separation occurring in the shopping trouble model has several characteristic features. One is that upper and lower-class engage in qualitatively different strategies. There are as many different pure strategies as there are upper-class agents, one for each, but only one mixed strategy for the entire lower-class. Individualism is lost when becoming a member of the masses, to put it colloquially. Alternatively, one may view the common mixed strategy played by the lower-class as an atypical group-level trait, namely one that does not come with an improved Darwinian fitness [39]. For an understanding, we note that envy enters the shopping trouble model as $\varepsilon p^\alpha(q_i) \log(R^\alpha/\bar{R})$, which implies that the current probability $p^\alpha(q_i)$ to select a given quality tends to be suppressed when $R^\alpha < \bar{R}$. Envy has a self-reinforcing effect when the individual reward $R^\alpha$ is in contrast not smaller, but larger than the population average. This argument explains why agents with modest/high rewards play mixed/pure strategies.

Evolutionary stable strategies can have different rewards only when their supports are not identical, which becomes increasingly difficult with the continuous increase in the number of mixed strategies that is observed during forced cooperation with raising levels of envy, see figure 5. Policies merge once the phase space for the support of distinct mixed strategies runs out and a single mixed lower-class

strategy remains. Class stratification corresponds from this perspective to a strategy merging transition, producing in consequence an atypical group-level trait.

A second feature characterizing class stratification is universality, namely that the underlying utility function $v(q_i)$ affects the Nash equilibrium exclusively through the fraction $f_L$ of lower-class agents. An interesting corollary is that is does not really matter which options the upper-class selects, as the reward, and consequently also the monetary income, remains unaffected. One could call this freedom the luxury of choice of being rich. Upper-class strategies tend to cluster, nevertheless, around the maximum of the underlying utility function (compare figure 3), which is however a purely dynamic effect. Policies that prefer qualities with large $v(q_i)$ have increased growth rates while iterating towards stationarity.

Beyond its original interpretation as a competitive shopping model, one can view the shopping trouble model as a basic model for competition for scarce goods, in particular in a social context. The qualities $q_i$ would correspond in this setting either to distinct social positions or to job opportunities, with the bare utility $v(q_i)$ encoding, respectively, social status and salaries. It is presently unclear to what extent, and if at all, human societies can be described in a first approximation by the shopping trouble model. In that case, Western societies are presumable in the phase denoted here as forced cooperation, with varying distances to the class stratification transition. A transition to the stratified phase would be equivalent to a major socio-cultural paradigm shift [40], such as the possible incipient dynamic instability of modern democracies due to growing mismatch between the built-in time delays, the election cycle and the accelerating pace of political opinion dynamics [41]. This is a somewhat worrisome outlook, given that the repercussions of envy are amplified, as shown in figure 6, when societies become more and more competitive: a possible ongoing development [42].

The stratified phase found in the shopping trouble model is the result of a self-organizing process, with the consequence that it has universal properties that can be controlled only indirectly by external influences. Policy-makers lose part of their tools when a society class separates. Class-stratified societies are in this sense intrinsically resistant to external influences. Overall our results show that envy tends to cement class differences, instead of softening them. It may be tempting for people at the bottom to compare what they have with the riches of the top, but it is actually counterproductive.

Data accessibility. This article has no additional data.

Competing interests. This article has no competing interest.

Acknowledgements. The author thanks Daniel Gros for discussions and Roser Valenti for valuable suggestions regarding the manuscript.

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
