## [Reviewer comments · Royal Society Open Science]

Review History

RSOS-200411.R0 (Original submission)

Review form: Reviewer 1

Is the manuscript scientifically sound in its present form?

Yes

Are the interpretations and conclusions justified by the results?

Yes

Is the language acceptable?

Yes

Do you have any ethical concerns with this paper?

No

Have you any concerns about statistical analyses in this paper?

No

Recommendation?

Accept as is

Comments to the Author(s)

The manuscript "Self induced class stratification in competitive societies of agents: Nash stability in the presence of envy." by author Gros presents a model on the dynamical effects of envy in social dynamics and organisation that is interesting and to my knowledge new and therefore should be published.

It represents envy in a game theoretical setting of a so-called shopping trouble model in a delightful and creative way and observes a self-organised stratification of the toy model society as a function of sensible parameters. The observed ordering effects from this physics of complex adaptive system's perspective are interesting and the implicit claim of practical relevance via universality makes this a prime system for further discussion in the field of sociophysics. I like the model much. My personal caveat is that the model definition eq. 2.2 is very well suited for physicists --- but perhaps not so much as a minimalistic toy model for less mathematically inclined, and therefore will probably not turn viral in the sociology community right away. I do not have any ad hoc suggestion, though, other than encouraging the author to think about more realistic, yet at the same time simpler payoff functions or scenarios in a second paper.

This paper should be published as is (after correcting the mixed up variables alpha and epsilon in caption 2) and will definitely motivate interesting follow-up research.

Review form: Reviewer 2

Is the manuscript scientifically sound in its present form?

Yes

Are the interpretations and conclusions justified by the results?

Yes

Is the language acceptable?

Yes

Do you have any ethical concerns with this paper?

No

Have you any concerns about statistical analyses in this paper?

No

Recommendation?

Major revision is needed (please make suggestions in comments)

Comments to the Author(s)

Summary of the paper:

The author studies increasing levels of envy in a game-theoretic setting and shows via numerical simulations that this in itself causes a separation of society into a lower and an upper class. Further findings include that class stratification is Nash stable and strict when members of the same class receive the same rewards; upper-class members play only pure strategies, whereas lower-class members play the same mixed strategy; the fraction of upper-class members decreases progressively with larger levels of envy until just a single upper class agent is left. The performed numerical simulations are based on a reference model, the so-called "shopping trouble model." The author also discusses the impact of his findings for human societies.

Recommendation:

The findings of this paper, which by and large is well written (but see also the listed points of criticism and the minor suggestions for improvement below), are certainly interesting and novel and deserve publication. My main point of criticism is that the paper seems to be written by an expert for other experts only. It is not well accessible for a wider audience. Therefore, I recommend to accept this paper subject to a major revision.

Some points of criticism:

Many of the notions and concepts used come without explanation or definition, so for someone working in a different special area it is really hard to understand the results of this paper. For example, it should first be said what specific type of game-theoretic model is used and how it is formally defined, including utility functions and their domain and range.

The notion of envy is informally explained on page 2 but should also be formally defined.

The same applies to the notions of Nash state, Nash stability, and so on. For example, in the setting of hedonic games (or, more generally, coalition formation games), Nash stability means that no player prefers moving to another coalition. But since we here are in a non-cooperative game (I guess), I assume that Nash stability means something else here. However, I cannot understand what it really means if no formal definition is given.

Regarding the (shopping trouble) model explained on pages 3--4, in particular through the displayed formulas (2.1), (2.2), and (2.3), it is unclear whether the author introduces this model here or whether it has been already proposed in the literature and he merely keeps studying it. Also, it would be nice to compare this specific model with other (perhaps related, perhaps similar) models occurring in the literature and to explain what the main advantages of the present model are.

Minor suggestions for improvement:

page 1, abstract:

"into a lower- and an upper class": Delete the hyphen.

page 1, bottom:

In the last line, after "selecting the same option are" there is some empty space that should be filled.

page 2, line 19:

"the problem of allocating multiple types of divisible or indivisible goods, like apples, banana and kiwis":
It is unclear whether you consider apples, banana and kiwis as divisible or indivisible goods.

page 2, line 29:

independently whether => independently of whether

page 2, line 31:

What does "f.i." mean? It is not very common.

page 2, line 31:

"jackpot": A jackpot typically contains everything, i.e., the accumulation of unwon rewards. So I don't know what you mean by "second best option".

page 3, line 52:
Fix "support of a strategies".

page 4, line 6:
where have defined => where we have defined

page 4, line 28:

a functionality that is equivalently =>
a functionality that is equivalent

page 7, line 57:
"we present to this extend": Do you mean "to this end" or "to this extent"?

page 8, line 28:
in contrast to the lower- and upper class rewards =>
in contrast to the lower- and upper-class rewards

page 9, line 52:
Delete the comma at the end of Eq. (3.5).

page 10, line 48:
This sentence reads better if you shift "numerically" to the end of the sentence. In the next sentence, "Shown is" looks also a bit misplaced.

Decision letter (RSOS-200411.R0)

Dear Dr Gros

On behalf of the Editors, I am pleased to inform you that your Manuscript RSOS-200411 entitled "Self induced class stratification in competitive societies of agents: Nash stability in the presence of envy" has been accepted for publication in Royal Society Open Science subject to minor revision in accordance with the referee suggestions. Please find the referees' comments at the end of this email.

The reviewers and handling editors have recommended publication, but also suggest some minor revisions to your manuscript. Therefore, I invite you to respond to the comments and revise your manuscript.

- Ethics statement

- Data accessibility

It is a condition of publication that all supporting data are made available either as supplementary information or preferably in a suitable permanent repository. The data accessibility section should state where the article's supporting data can be accessed. This section

should also include details, where possible of where to access other relevant research materials such as statistical tools, protocols, software etc can be accessed. If the data has been deposited in an external repository this section should list the database, accession number and link to the DOI for all data from the article that has been made publicly available. Data sets that have been deposited in an external repository and have a DOI should also be appropriately cited in the manuscript and included in the reference list.

If you wish to submit your supporting data or code to Dryad (<http://datadryad.org/>), or modify your current submission to dryad, please use the following link:
<http://datadryad.org/submit?journalID=RSOS&manu=RSOS-200411>

- **Competing interests**

- **Authors' contributions**

- **Acknowledgements**

- **Funding statement**

Because the schedule for publication is very tight, it is a condition of publication that you submit the revised version of your manuscript before 28-May-2020. Please note that the revision deadline will expire at 00.00am on this date. If you do not think you will be able to meet this date please let me know immediately.

If your manuscript is newly submitted and subsequently accepted for publication, you will be asked to pay the article processing charge, unless you request a waiver and this is approved by Royal Society Publishing. You can find out more about the charges at <https://royalsocietypublishing.org/rsos/charges>. Should you have any queries, please contact openscience@royalsociety.org.

on behalf of Professor Tim Rogers (Associate Editor) and Mark Chaplain (Subject Editor)
openscience@royalsociety.org

Associate Editor Comments to Author (Professor Tim Rogers):

Comments to the Author:

Please implement the changes recommended by both referees. In particular, care should be taken to ensure the manuscript is self contained and all necessary terms are precisely defined, or suitable references given.

Reviewer comments to Author:

Reviewer: 1

Comments to the Author(s)

The manuscript "Self induced class stratification in competitive societies of agents: Nash stability in the presence of envy." by author Gros presents a model on the dynamical effects of envy in social dynamics and organisation that is interesting and to my knowledge new and therefore should be published.

It represents envy in a game theoretical setting of a so-called shopping trouble model in a delightful and creative way and observes a self-organised stratification of the toy model society as a function of sensible parameters. The observed ordering effects from this physics of complex adaptive system's perspective are interesting and the implicit claim of practical relevance via universality makes this a prime system for further discussion in the field of sociophysics. I like the model much. My personal caveat is that the model definition eq. 2.2 is very well suited for physicists --- but perhaps not so much as a minimalistic toy model for less mathematically inclined, and therefore will probably not turn viral in the sociology community right away. I do not have any ad hoc suggestion, though, other than encouraging the author to think about more realistic, yet at the same time simpler payoff functions or scenarios in a second paper.

This paper should be published as is (after correcting the mixed up variables alpha and epsilon in caption 2) and will definitely motivate interesting follow-up research.

Reviewer: 2

Comments to the Author(s)

Summary of the paper:

The author studies increasing levels of envy in a game-theoretic setting and shows via numerical simulations that this in itself causes a separation of society into a lower and an upper class.

Further findings include that class stratification is Nash stable and strict when members of the same class receive the same rewards; upper-class members play only pure strategies, whereas lower-class members play the same mixed strategy; the fraction of upper-class members decreases progressively with larger levels of envy until just a single upper class agent is left. The performed numerical simulations are based on a reference model, the so-called "shopping trouble model." The author also discusses the impact of his findings for human societies.

Recommendation:

The findings of this paper, which by and large is well written (but see also the listed points of criticism and the minor suggestions for improvement below), are certainly interesting and novel and deserve publication. My main point of criticism is that the paper seems to be written by an expert for other experts only. It is not well accessible for a wider audience. Therefore, I recommend to accept this paper subject to a major revision.

Some points of criticism:

Many of the notions and concepts used come without explanation or definition, so for someone working in a different special area it is really hard to understand the results of this paper. For example, it should first be said what specific type of game-theoretic model is used and how it is formally defined, including utility functions and their domain and range.

The notion of envy is informally explained on page 2 but should also be formally defined.

The same applies to the notions of Nash state, Nash stability, and so on. For example, in the setting of hedonic games (or, more generally, coalition formation games), Nash stability means that no player prefers moving to another coalition. But since we here are in a non-cooperative game (I guess), I assume that Nash stability means something else here. However, I cannot understand what it really means if no formal definition is given.

Regarding the (shopping trouble) model explained on pages 3--4, in particular through the displayed formulas (2.1), (2.2), and (2.3), it is unclear whether the author introduces this model here or whether it has been already proposed in the literature and he merely keeps studying it. Also, it would be nice to compare this specific model with other (perhaps related, perhaps similar) models occurring in the literature and to explain what the main advantages of the present model are.

Minor suggestions for improvement:

page 1, abstract:

"into a lower- and an upper class": Delete the hyphen.

page 1, bottom:

In the last line, after "selecting the same option are" there is some empty space that should be filled.

page 2, line 19:

"the problem of allocating multiple types of divisible or indivisible goods, like apples, banana and kiwis":
It is unclear whether you consider apples, banana and kiwis as divisible or indivisible goods.

page 2, line 29:

independently whether => independently of whether

page 2, line 31:

What does "f.i." mean? It is not very common.

page 2, line 31:

"jackpot": A jackpot typically contains everything, i.e., the accumulation of unwon rewards. So I don't know what you mean by "second best option".

page 3, line 52:
Fix "support of a strategies".

page 4, line 6:
where have defined => where we have defined

page 4, line 28:

a functionality that is equivalently =>
a functionality that is equivalent

page 7, line 57:
"we present to this extend": Do you mean "to this end" or "to this extent"?

page 8, line 28:
in contrast to the lower- and upper class rewards =>
in contrast to the lower- and upper-class rewards

page 9, line 52:
Delete the comma at the end of Eq. (3.5).

page 10, line 48:
This sentence reads better if you shift "numerically" to the end of the sentence. In the next sentence, "Shown is" looks also a bit misplaced.

Author's Response to Decision Letter for (RSOS-200411.R0)

See Appendix A.

Decision letter (RSOS-200411.R1)

Dear Dr Gros,

It is a pleasure to accept your manuscript entitled "Self induced class stratification in competitive societies of agents: Nash stability in the presence of envy" in its current form for publication in Royal Society Open Science.

Due to rapid publication and an extremely tight schedule, if comments are not received, your paper may experience a delay in publication. Royal Society Open Science operates under a continuous publication model. Your article will be published straight into the next open issue and

this will be the final version of the paper. As such, it can be cited immediately by other researchers. As the issue version of your paper will be the only version to be published I would advise you to check your proofs thoroughly as changes cannot be made once the paper is published.

on behalf of Professor Tim Rogers (Associate Editor) and Mark Chaplain (Subject Editor)
openscience@royalsociety.org

Appendix A

Letter to the editor

Dear Anita Kristiansen, Editorial Coordinator

We are happy that the reviewers support our research and that both referees suggest to extend the discussion of the terminology used. We have been aware that it would have helped to explain the terminology in a dedicated section. As this is somewhat unusual, we did not add this section in the initial version of the manuscript. We have done so now, following the suggestions of the referees, in particular of referee 2. The new section '4.5 Terminology', at the end of the methods section, is referred-to at the end of the introduction.

A small number of additional references and other minor corrections have been included.

We are looking forward to the see article published.

Yours
Claudius Gros

Response to the comments of Reviewer 1

This paper should be published as is (after correcting the mixed up variables
alpha and epsilon in caption 2) and will definitely motivate interesting
follow-up research.

We thank the reviewer for her/his support and pointing out that alpha and epsilon have been mixed in the caption Fig. 2, which we corrected

Response to the comments of Reviewer 2

We thank the reviewer for their support and for the valuable suggestions, which we implemented in full.

Many of the notions and concepts used come without explanation or definition,
so for someone working in a different special area it is really hard to
understand the results of this paper. For example, it should first be said
what specific type of game-theoretic model is used and how it is formally
defined, including utility functions and their domain and range.
The notion of envy is informally explained on page 2 but should also be
formally defined.
The same applies to the notions of Nash state, Nash stability, and so on. For
example, in the setting of hedonic games (or, more generally, coalition
formation games), Nash stability means that no player prefers moving to another
coalition. But since we here are in a non-cooperative game (I guess), I assume
that Nash stability means something else here. However, I cannot understand
what it really means if no formal definition is given.

We introduced a new section, 4.5 Terminology, in which we present, as we hope, an exhaustive list of formal definitions.

Regarding the (shopping trouble) model explained on pages 3--4, in particular # through the displayed formulas (2.1), (2.2), and (2.3), it is unclear whether # the author introduces this model here of whether it has been already proposed # in the literature and he merely keeps studying it. Also, it would be nice to # compare this specific model with other (perhaps related, perhaps similar) # models occurring in the literature and to explain what the main advantages of # the present model are.

True, this was never stated explicitly. In Sect. 2 we now point out that the shopping trouble model is introduced in this paper, see ``The bare formulation of the here introduced shopping trouble game, ...''

We also introduced a new paragraph somewhat below Eq.(2.2), starting with ``The utility function (2.2) of the shopping trouble model'', where the relation of our model with status seeking games is discussed.

Minor suggestions for improvement:
Thank you very much for these suggestions, which we implemented.

In the last line, after "selecting the same option are" there is some # empty space that should be filled.

This is actually a particularity of the Royal Society format, which will be taken into account by the production editor.

"the problem of allocating multiple types of divisible # or indivisible goods, like apples, banana and kiwis": # It is unclear whether you consider apples, banana and kiwis as # divisible or indivisible goods.
True, reformulated.

What does "f.i." mean? It is not very common.

For instance. Has been written out in the revised version.

"jackpot": A jackpot typically contains everything, i.e., the # accumulation of unwon rewards. So I don't know what you mean # by "second best option".

This expression was, admittedly, too colloquial. The sentence in question has been rewritten.

Fix "support of a strategies".

Done.

where have defined => where we have defined

Done.

a functionality that is equivalently =>
a functionality that is equivalent

Actually, we believe that equivalently is fine here.

"we present to this extend": Do you mean "to this end" or "to this extent"?

The first, corrected.

in contrast to the lower- and upper class rewards =>

in contrast to the lower- and upper-class rewards

Done.

Delete the comma at the end of Eq. (3.5).

Done.

This sentence reads better if you shift "numerically" to the end of the

sentence. In the next sentence, "Shown is" looks also a bit misplaced.

Corrected / reformulated. Thanks.